# Axon injury triggers EFA-6 mediated destabilization of axonal microtubules via TACC and doublecortin like kinase

Lizhen Chen[1,2], Marian Chuang[1], Thijs Koorman[3], Mike Boxem[3], Yishi Jin[1,2,4*], Andrew D Chisholm[1*]

[1]Neurobiology Section, Division of Biological Sciences, University of California, San Diego, La Jolla, United States; [2]Howard Hughes Medical Institute, University of California, San Diego, La Jolla, United States; [3]Department of Biology, Utrecht University, Utrecht, Netherlands; [4]Department of Cellular and Molecular Medicine, University of California, San Diego School of Medicine, La Jolla, United States

**Abstract** Axon injury triggers a series of changes in the axonal cytoskeleton that are prerequisites for effective axon regeneration. In *Caenorhabditis elegans* the signaling protein Exchange Factor for ARF-6 (EFA-6) is a potent intrinsic inhibitor of axon regrowth. Here we show that axon injury triggers rapid EFA-6-dependent inhibition of axonal microtubule (MT) dynamics, concomitant with relocalization of EFA-6. EFA-6 relocalization and axon regrowth inhibition require a conserved 18-aa motif in its otherwise intrinsically disordered N-terminal domain. The EFA-6 N-terminus binds the MT-associated proteins TAC-1/Transforming-Acidic-Coiled-Coil, and ZYG-8/Doublecortin-Like-Kinase, both of which are required for regenerative growth cone formation, and which act downstream of EFA-6. After injury TAC-1 and EFA-6 transiently relocalize to sites marked by the MT minus end binding protein PTRN-1/Patronin. We propose that EFA-6 acts as a bifunctional injury-responsive regulator of axonal MT dynamics, acting at the cell cortex in the steady state and at MT minus ends after injury.

*For correspondence:
yijin@ucsd.edu (YJ);
chisholm@ucsd.edu (ADC)

**Competing interests:** The authors declare that no competing interests exist.

## Introduction

In mature nervous systems axons regenerate poorly after injury, leading to permanent functional deficits. Both the nature of the extracellular environment and the intrinsic growth competence of the neuron contribute to the extent of axon regeneration (*Case and Tessier-Lavigne, 2005*). The mammalian central nervous system (CNS) expresses a variety of environmental regeneration inhibitory factors, including myelin-associated proteins, chondroitin sulfate proteoglycans and glial scar tissue that functions as a physical barrier (*Schwab, 2004*; *Silver and Miller, 2004*). However genetic removal of these inhibitory factors results in only limited improvement in regeneration of severed axons (*Lee et al., 2009*, *2010*). Recent studies have strongly supported the importance of cell-intrinsic determinants in axon regeneration. Loss of function in cell-intrinsic growth inhibitors such as Phosphatase and Tensin homolog, PTEN, and Suppressor Of Cytokine Signaling-3, SOCS3, can dramatically improve axon regrowth even in the inhibitory CNS environment (*Park et al., 2008*; *Sun et al., 2011*). Genetic and pharmacological manipulation of cell autonomous signaling pathways can dramatically improve regrowth of severed axons in various injury paradigms (*Moore et al., 2009*; *Hellal et al., 2011*; *Sengottuvel et al., 2011*; *Shin et al., 2012*; *Watkins et al., 2013*; *Ruschel et al., 2015*).

During developmental axon outgrowth and in regenerative regrowth of mature neurons, the formation and extension of growth cones involve extensive remodeling of the microtubule (MT) cytoskeleton (*Bradke et al., 2012*; *Chisholm, 2013*). Cellular compartments undergoing rapid

**eLife digest** In the nervous system, cells called neurons carry information around the body. These cells have long thin projections called axons that allow the information to pass very quickly along the cell to junctions with other neurons. Neurons in adult mammals are limited in their ability to regenerate, so any damage to axons, for example, due to a stroke or a brain injury, tends to be permanent. Therefore, an important goal in neuroscience research is to discover the genes and proteins that are involved in regenerating axons as this may make it possible to develop new therapies.

An internal scaffold called the cytoskeleton supports the three-dimensional shape of the axons. Changes in the cytoskeleton are required to allow neurons to regenerate axons after injury, and drugs that stabilize filaments called microtubules in the cytoskeleton can promote these changes. Chen et al. used a technique called laser microsurgery to sever individual axons in a roundworm known as *C. elegans* and then observed whether these axons could regenerate. The experiments reveal that a protein called EFA-6 blocks the regeneration of neurons by preventing rearrangements in the cytoskeleton.

EFA-6 is normally found at the membrane that surrounds the neuron. However, Chen et al. show that when the axon is damaged, this protein rapidly moves to areas near the ends of microtubule filaments. EFA-6 interacts with two other proteins that are associated with microtubules and are required for axons to be able to regenerate. Chen et al.'s findings demonstrate that several proteins that regulate microtubule filaments play a key role in regenerating axons. All three of these proteins are found in humans and other animals so they have the potential to be targeted by drug therapies in future. The next challenge is to understand the details of how EFA-6 activity is affected by axon injury, and how this alters the cytoskeleton.

morphological changes, such as axonal growth cones, are enriched in dynamic MTs (*Suter et al., 2004*), while mature axons or dendrites contain predominantly stabilized MTs (*Baas et al., 1993*). When an axon is injured, MTs are locally disassembled or severed, potentially creating free plus ends for new MT polymerization. Subsequently, the number of growing MTs increases, followed by more persistent MT growth correlated with formation of regenerative growth cone and axon extension (*Erez and Spira, 2008*; *Ghosh-Roy et al., 2012*). Complete removal of an axon also leads to dramatic upregulation of MT dynamics in the soma and dendrites (*Stone et al., 2010*). MT disorganization contributes to dystrophic end bulb formation after injury in CNS (*Ertürk et al., 2007*). Moderate stabilization of MT dynamics by Taxol or other MT stabilizers can promote axon regrowth in vitro and in the mammalian CNS (*Usher et al., 2010*; *Hellal et al., 2011*; *Sengottuvel et al., 2011*; *Ruschel et al., 2015*); the effects of Taxol in vivo have been partly replicated (*Popovich et al., 2014*; *Ruschel et al., 2015*). Thus, there is a critical need to define the endogenous regulators of MTs after injury.

In a large-scale screen for genes affecting adult axon regeneration in *Caenorhabditis elegans*, we identified Exchange Factor for ARF-6 (EFA-6) as an intrinsic inhibitor of axon regrowth (*Chen et al., 2011*). The EFA-6/EFA6 protein family is conserved from yeast to mammals, and is defined by its Sec7 domain, which confers guanine exchange factor (GEF) activity for Arf6 GTPases (*Franco et al., 1999*). Unexpectedly, the regrowth-inhibitory function of EFA-6 is independent of its GEF activity, and instead is mediated by its N-terminal domain (*Chen et al., 2011*). The EFA-6 N-terminal domain inhibits MT growth at the cell cortex of *C. elegans* embryos via a conserved motif of 18 amino acids (*O'Rourke et al., 2010*). Nonetheless, the mechanism by which EFA-6 regulates MT dynamics is unknown.

Here we reveal that axon injury triggers rapid and transient relocalization of EFA-6, concomitant with an initial downregulation of axonal MT dynamics. The N-terminal 18-aa motif is required for injury-induced relocalization and for inhibition of axonal MT growth. We show that the EFA-6 N-terminal domain interacts with MT associated proteins TAC-1, a member of the transforming acidic coiled-coil (TACC) family, and ZYG-8, an ortholog of doublecortin-like kinase (DCLK). TAC-1 and ZYG-8 are required for initiation of axon regeneration, and their overexpression can promote regrowth. We further show that injury triggers relocalization of EFA-6 and TAC-1 to sites overlapping with the MT minus end binding protein Patronin/PTRN-1. We propose that EFA-6 is a bifunctional injury-responsive regulator of MT dynamics, acting at the cell cortex in the steady state and at MT minus ends after axon injury.

## Results

### Axon injury triggers redistribution of EFA-6

In the one-cell stage embryo EFA-6 localizes to the plasma membrane via its C-terminal PH (pleckstrin homology) domain, and this plasma membrane localization of EFA-6 is necessary for it to inhibit cortical MT growth (*O'Rourke et al., 2010*). To determine the subcellular location of EFA-6 in neurons, we expressed a series of GFP-tagged EFA-6 fusion proteins. Full-length EFA-6 tagged with GFP at the N- or C-termini, expressed at a range of concentrations, localized to the plasma membrane of the soma and processes of neurons (*Figure 1A,B*, *Figure 1—figure supplement 1A*); deletion of the PH domain (FLΔPH) resulted in cytosolic localization (*Figure 1C*, upper panel). The first 150 residues of the EFA-6 N-terminus (N150), expression of which inhibits axon regrowth (*Chen et al., 2011*), was localized to the cytosol similarly to FLΔPH (*Figure 1C,D*). Conversely, EFA-6 proteins lacking the N-terminal 150 amino acids (FLΔN150) showed plasma membrane localization resembling that of full-length EFA-6 (*Figure 1B,E*).

We next examined how axon injury affected EFA-6 localization. Within seconds of laser axotomy of the PLM axon, GFP::EFA-6FL redistributed from a generally even plasma membrane localization to more discrete puncta (*Figure 1B*, *Video 1*). This relocalization was also observed when GFP was tagged to the C terminus or in the middle of EFA-6 (*Figure 1—figure supplement 1B*), and did not require the PH domain (*Figure 1C*). GFP::EFA-6N150, although not membrane associated, also became punctate after injury, whereas GFP::EFA-6FLΔN150 did not relocalize after axon injury (*Figure 1D,E*). Full-length EFA-6 and the N terminal domain (N150) appear to relocalize to the same puncta in response to injury, as shown by co-expressing EFA-6FL::GFP and EFA-6N150::mKate2 (*Figure 1F*, *Figure 1—figure supplement 2A*). mKate2::EFA-6N150Δ18 (which does not relocalize, see below) did not co-localize with EFA-6FL::GFP after injury, suggesting EFA-6 puncta are not due to non-specific aggregation of proteins after injury (*Figure 1—figure supplement 2B*). The region of relocalized EFA-6FL or EFA-6N150 expanded bidirectionally from the injury site at ~8 μm s$^{-1}$, into the soma and dendrite (*Figure 1—figure supplement 2C*, *Video 1*). The density of injury-triggered puncta of EFA-6FL and EFA-6N150 gradually decreased over the next 20–60 min (*Figure 1B*, *Figure 1—figure supplement 3A–C*). We observed similar re-localization of GFP::EFA-6 after injury of motor neuron axons (*Figure 1—figure supplement 3D,E*), as well as when injury was delivered to the soma, distal axon, or posterior processes of mechanosensory neurons (*Figure 1—figure supplement 3F*). These results indicate that injury-triggered EFA-6 relocalization occurs in multiple neuron types, irrespective of the site of injury and independent of the Sec7 or PH domains. The N-terminal 1–70 aa (N70) was the smallest fragment tested that displayed relocalization after injury (*Figure 1I*). We tested a variety of other neuronal proteins, including the MT plus-end binding proteins EBP-1 and EBP-2, KLP-7/kinesin-13 (*Chen et al., 2011*; *Ghosh-Roy et al., 2012*), ARF-6 (the presumed substrate for EFA-6's GEF activity), SAX-3 (transmembrane receptor) and synaptic vesicle proteins (SNB-1/synaptobrevin, RAB-3/GTPase, UNC-57/endophilin), and found that none showed similar relocalization after axon injury (*Figure 1—figure supplement 3G*, and data not shown). Thus, injury-triggered relocalization is specific to EFA-6.

The punctate distribution of EFA-6 after injury suggested that EFA-6 might become sequestered to a subcellular compartment. To address whether injury alters EFA-6 mobility within the cell we performed FRAP (Fluorescence Recovery After Photobleaching). In uninjured neurons, EFA-6FL::GFP recovered with $t_{1/2} = 4$ s and an immobile fraction of 25%. In contrast, EFA-6FL::GFP puncta 2 min after injury showed dramatically reduced recovery, with >85% of the protein in the immobile fraction (*Figure 2A,B*). We were not able to calculate $t_{1/2}$ due to the extremely low recovery rate. By 1 hr after injury, EFA-6FL::GFP partially returned to its steady state localization (*Figure 1B*), and its recovery rate was partially restored, with an immobile fraction of 50% (*Figure 2A,B*). GFP::EFA-6N150 gave similar FRAP results (not shown). This analysis suggests that after injury EFA-6 is sequestered to subcellular structures.

### The ability of EFA-6 fragments to inhibit developmental axon outgrowth and regrowth after axon injury correlates with their localization

*efa-6(lf)* mutants display significantly increased PLM axon regrowth following laser axotomy in adults, as well as impenetrant developmental overshooting of PLM axons; conversely, overexpression of

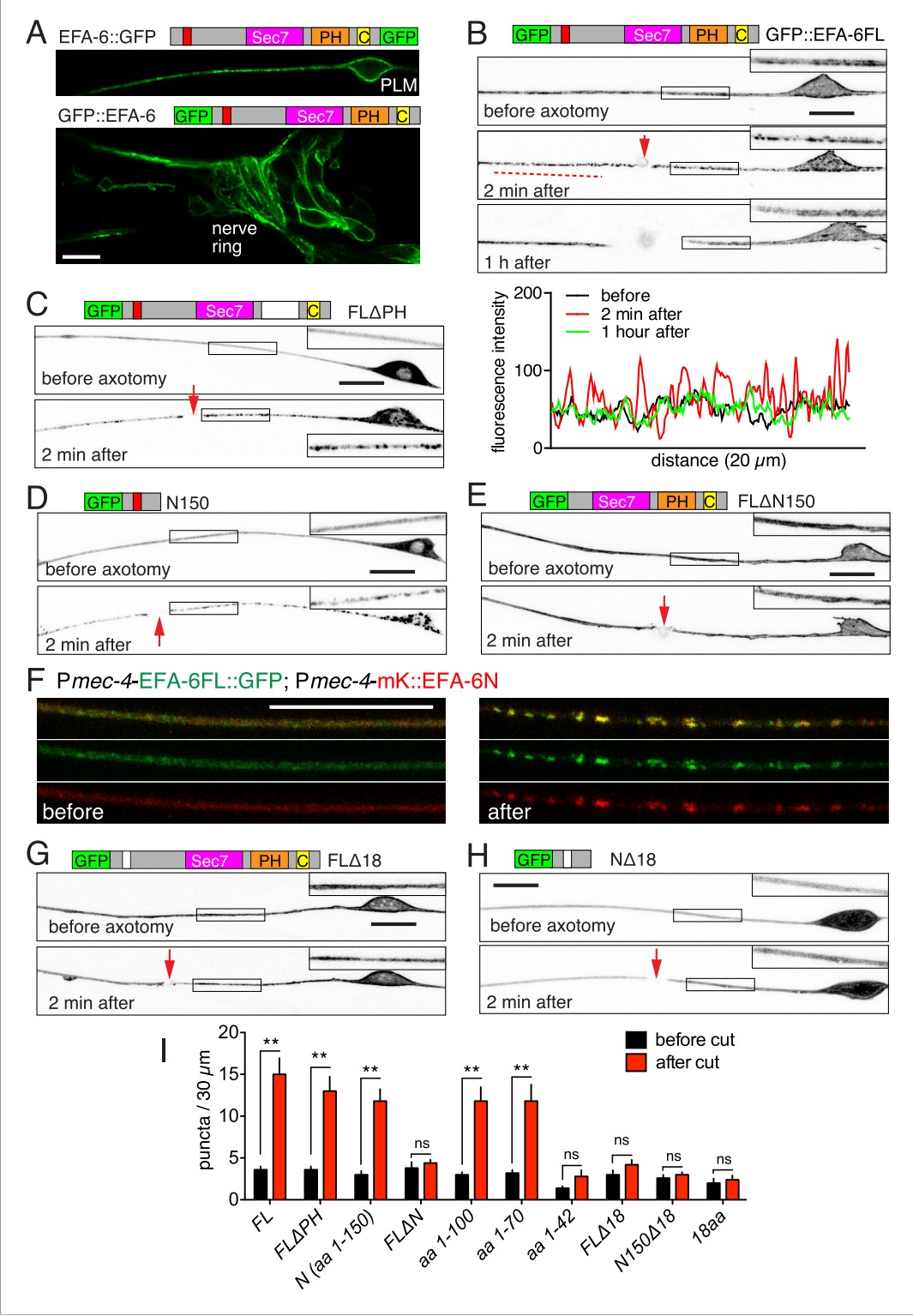

**Figure 1**. Axon injury triggers rapid relocalization of Exchange Factor for ARF-6 (EFA-6), mediated by its N-terminal domain. (**A**) Single focal plane images of PLM (top) and nerve ring (bottom) showing membrane localization of EFA-6. Transgenes: P*mec-4*-EFA-6::GFP (*juEx6467*) (top) and P*rgef-1*-GFP::EFA-6 (*juEx6374*) (bottom). (**B**) Localization of full length EFA-6 (P*mec-4*-GFP::EFA-6, *juEx6160*) before, 2 min after, and 1 hr post axotomy. Projections of confocal z stacks, inverted grayscale; enlargements in inserts. Bottom, fluorescence intensity along line scan. (**C–E**) Localization of GFP::EFA-6 fusion protein lacking the PH domain (FLΔPH) (P*mec-4*-GFP::EFA-6 FLΔPH, *juEx6453*), EFA-6 N-

*Figure 1. continued on next page*

*Figure 1. Continued*

terminal 150 aa (N150) (P*mec-4*-GFP::EFA-6N150, *juEx3531*), and EFA-6 lacking the N-terminus (FLΔN150) (P*mec-4*-GFP::EFA-6FLΔN150, *juEx6154*). (**F**) Colocalization of EFA-6FL and EFA-6N150 puncta after injury. Localization of EFA-6FL::GFP and EFA-6N150::mKate2 (*juEx6522*) in PLM before and after axotomy. EFA-6 full length protein and N terminus relocalize to overlapping puncta. (**G**, **H**) Requirement for the 18-aa motif for relocalization of EFA-6FL and EFA-6N150. Localization of GFP::EFA-6FLΔ18aa (*juEx6156*), and GFP::EFA-6N150Δ18aa (*juEx3535*) in touch neurons before and 2 min after axotomy. (**I**) Quantitation of puncta before and 2 min after injury in axons expressing different EFA-6 fragments. Statistics, one-way ANOVA with Bonferroni post test; n = 5 for each bar; **p < 0.01, ns, not significant. Scale, 10 μm.

The following figure supplements are available for figure 1:

**Figure supplement 1**. Injury induced relocalization of EFA-6 is independent of expression level or location of tag.

**Figure supplement 2**. Full-length EFA-6 and N terminus relocalize to the same puncta after injury.

**Figure supplement 3**. Injury induced relocalization of EFA-6.

EFA-6 strongly inhibits axon regrowth after axotomy, and causes premature truncation of PLM axon growth ('undershooting') in development (*Chen et al., 2011*) (*Figure 2C,D*, *Figure 2—figure supplement 1A*). We therefore tested if the capacity of EFA-6 protein fragments to relocalize correlated with their effects on PLM development and regrowth. We found that overexpression of the EFA-6 N terminal fragments that displayed injury-induced relocalization (N150, N100, N70) all caused axons of PLM neurons to undershoot (*Figure 2C*, *Figure 2—figure supplement 1A*; *Table 1*). We performed axotomy on those axons that exhibited normal morphology in L4 larvae and found that these axons all showed significantly reduced regrowth (*Figure 2D*, *Figure 2—figure supplement 1B*). In contrast, overexpression of smaller fragments (N24, N42 or N[43–70 aa]) of the EFA-6 N-terminus that did not relocalize in response to axotomy did not significantly inhibit PLM outgrowth or regrowth (*Figure 2C,D* and *Table 1*). Interestingly, overexpression of EFA-6FLΔN150, but not of EFA-6FL, caused PLM overshooting, suggesting the C-terminal region of EFA-6 may inhibit the activity of the N-terminus. Overexpression of EFA-6FL strongly inhibits axonal MT dynamics, as assessed using the EBP-GFP assay for growing MT plus ends (*Chen et al., 2011*). Using live imaging and quantitative kymograph analysis of EBP-2::GFP comets, we found that only those EFA-6 fragments displaying injury-induced relocalization also inhibited steady-state axonal MT dynamics when overexpressed (*Figure 2E*, *Figure 2—figure supplement 1C*). These observations suggest the ability of EFA-6 protein fragments to inhibit axon growth and MT dynamics closely correlates with their ability to relocalize in response to injury.

## EFA-6 function and relocalization require a conserved 18 aa motif in the N terminus

All EFA6 family members contain extended N-termini of low sequence complexity and lack defined sequence motifs (*Figure 2—figure supplement 2A*). The sole known region of sequence conservation among N-termini of some EFA6 family members is a motif of 18 aa (residues 25–42 in *C. elegans* EFA-6), necessary and partially sufficient for EFA-6's effects on cortical MT growth in the embryo (*O'Rourke et al., 2010*). Analysis of EFA-6/EFA6 N-termini using algorithms that predict natively disordered protein regions (*Xue et al., 2010*) indicates that the EFA-6 N-terminus has a high probability of protein disorder, with the exception of the 18 aa motif (*Figure 2—figure supplement 2B*).

To address the role of the 18 aa motif in EFA-6's function in neurons, we expressed constructs in

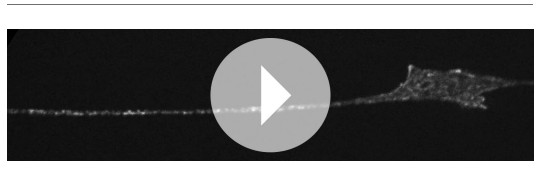

**Video 1.** Injury-induced GFP::EFA-6 relocalization in neurons (ALM). Transgene: P*mec-4*-GFP::EFA-6 (*juEx6160*). The video is 103 s, taken at 1 s/frame.

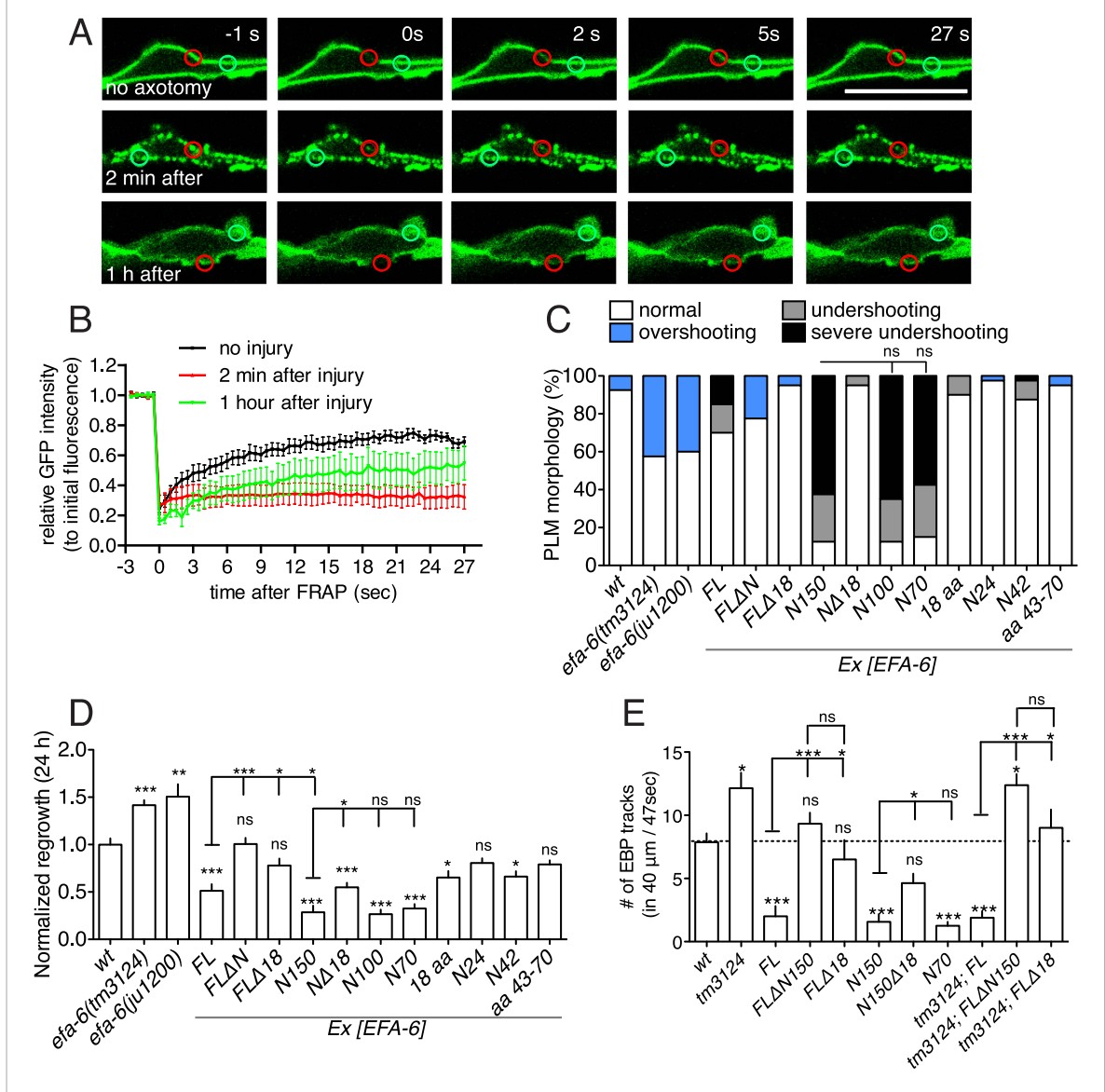

**Figure 2**. Injury-induced relocalization of EFA-6 correlates with its ability to regulate regrowth and microtubule (MT) dynamics. (**A**) FRAP of GFP::EFA-6 (*juEx6160*) before and after axon injury; regions of interest indicated by red circles; green circles were used to calibrate baseline fluorescence intensity. (**B**) Normalized average fluorescence intensity after FRAP. (**C**) PLM termination defects in *efa-6(lf)* mutants and EFA-6 overexpressing transgenic animals. n = 40 for each bar. See *Figure 2—figure supplement 1* for definitions of PLM overshooting and undershooting. (**D**) Normalized axon regrowth of *efa-6 (lf)* mutants and EFA-6 overexpressors. n ≥ 10. (**E**) Quantitation of EBP::GFP dynamics in intact axons from wt, *efa-6(tm3124)* and transgenic animals expressing different EFA-6 fragments under P*mec-4* promoter. n ≥ 10. Statistics, one-way ANOVA with Bonferroni post test; ***p < 0.001; **p < 0.01; *p < 0.05; ns, not significant.

The following figure supplements are available for figure 2:

**Figure supplement 1**. EFA-6 relocalization correlates with protein function in axon termination.

**Figure supplement 2**. The conserved 18-aa motif in the EFA-6 N-terminus is a region of local protein order.

which the 18 aa motif was deleted or mutated. We found that full length EFA-6 lacking the motif, GFP::EFA-6FLΔ18, localized to the plasma membrane but did not cause axonal developmental defects (*Figures 1G, 2C*). GFP::EFA-6N150 was diffuse in the cytosol and excluded from the nucleus (*Figure 1D*).

**Table 1**. Localization and function of EFA-6 protein fragments in PLM neurons

| EFA-6 proteins | GFP fusion protein localization | Injury-induced re-localization | Overexpression effect on regrowth | Overexpression effect on axon development |
|---|---|---|---|---|
| Full length (FL) | Cortical membrane | yes | 51.2% ± 7.1% *** | 30% undershooting |
| FLΔN150 | Cortical membrane | no | 100.5% ± 6.6% ns | 22.5% overshooting |
| N150 | Cytosolic | yes | 28.5% ± 7.1% *** | 87.5% undershooting |
| 18 aa | Cytosolic + nuclear | no | 65.3% ± 4.7% * | 10% mild undershooting |
| N150Δ18aa | Cytosolic + nuclear | no | 55.0% ± 4.7% *** | 5% mild undershooting |
| N150 (33–38A) | Cytosolic + nuclear | no | 57.6% ± 10.4% *** | 5% mild undershooting |
| N150 (25–32A) | Cytosolic + nuclear | no | 59% ± 5.7% *** | 7.5% mild undershooting |
| N150 S33A, D35A | Cytosolic + nuclear | no | 63% ± 7.8% ** | 7.5% mild undershooting |
| FLΔ18aa | Cortical membrane | no | 77.9% ± 7.3% ns | wt |
| N100 | Cytosolic | yes | 26.6% ± 4.7% *** | 87.5% undershooting |
| N70 | Cytosolic | yes | 32.7% ± 4.5% *** | 85% undershooting |
| N42 | Cytosolic + nuclear | no | 66.2% ± 5.7% * | 12.5% undershooting |
| N24 | Cytosolic + nuclear | no | 80.7% ± 5.0% ns | wt |

Mild and severe undershooting are defined as PLM termination anterior or posterior to the PVM soma respectively; 'undershooting' includes both mild and severe undershooting. See **Figure 2—figure supplement 1**.

Deletion of the 18 aa motif from the N-terminus (N150Δ18) abolished this nuclear exclusion, largely resembling free GFP (**Figure 1H**). Overexpression of EFA-6N150Δ18 also did not cause developmental abnormalities (**Figure 2C**, **Figure 2—figure supplement 1A**). Pan-neural overexpression of EFA-6N150, but not of EFA-6N150Δ18, caused aberrant locomotion (**Figure 2—figure supplement 1D,E**), suggesting that high levels of EFA-6 N-terminal domain perturb function of many neurons, in a manner dependent on the 18 aa motif. Moreover, mutating 2 or more residues within the 18 aa motif to alanines significantly reduced the activity of the N-terminus in multiple assays (**Table 1**), suggesting the sequence of the 18 aa motif is critical for the N-terminus to function. The 18 aa motif was essential for injury-triggered relocalization of EFA-6FL and EFA-6N150 (**Figure 1G,H**), as well as for their inhibitory effects in axon regrowth (**Figure 2D**). Expression of the 18 aa motif alone did not cause PLM developmental defects or confer injury-induced re-localization (**Figures 1I, 2C**), but mildly inhibited axon regrowth (**Figure 2D**), suggesting that injured axons are highly sensitive to the activity of this motif, but that the surrounding context is required for full activity of the N terminus.

Two *efa-6* loss of function alleles, *tm3124* and *ok3353*, delete genomic sequences that encode the Sec7 domain, and are predicted to cause frameshifts after the N-terminus (**Figure 2—figure supplement 2C**). Both mutations cause embryonic phenotypes similar to *efa-6* RNAi (**O'Rourke et al., 2010**) and display similarly enhanced axon regrowth (**Chen et al., 2011**). As these mutations do not delete the N-terminus, it is possible that truncated proteins might be produced in these mutants. We therefore generated a targeted deletion, *efa-6(ju1200)*, that removes the genomic sequences encoding the 18 aa motif (**Figure 2—figure supplement 2C**). The axon developmental and regrowth phenotypes of *efa-6(ju1200)* mutants were indistinguishable from those of *efa-6(tm3124)* (**Figure 2C,D**). In addition, single copy transgene expression of EFA-6N150 (*juSi86*) rescued the regeneration defects of both *efa-6 (tm3124)* and *efa-6(ju1200)* to similar degrees (**Figure 2—figure supplement 2D**), and also rescued developmental axon overgrowth (not shown). Thus, the increased axon regrowth of *efa-6* mutants reflects a complete loss of EFA-6 function, and the major activity of EFA-6 in axon growth resides in the N-terminus, dependent on the 18 aa motif. Below, we refer to *efa-6(tm3124)* as *efa-6(0)*.

## The EFA-6 N-terminus interacts with MT-associated proteins TAC-1 and ZYG-8

To understand how EFA-6 inhibits axon regeneration, we next searched for EFA-6 interacting proteins using yeast two-hybrid screening. We identified two strong interactors, the MT-associated proteins

(MAPs) TAC-1 and ZYG-8. ZYG-8 is the *C. elegans* ortholog of mammalian DCLK, defined by an N-terminal doublecortin domain and a C-terminal kinase domain (*Gönczy et al., 2001*). ZYG-8 is required for spindle positioning in embryos (*Gönczy et al., 2001*), and for normal axonal MT architecture in post-mitotic neurons (*Bellanger et al., 2012*). TAC-1 is the sole TACC protein in *C. elegans* (*Bellanger and Gönczy, 2003*; *Le Bot et al., 2003*; *Srayko et al., 2003*) and can form a complex with ZYG-8 to regulate MT assembly in embryos (*Bellanger et al., 2007*). In the yeast two-hybrid assay, we found that both TAC-1 and ZYG-8 interacted with EFA-6N150, dependent on the 18 aa motif (*Figure 3A*, *Figure 3—figure supplement 1*). TAC-1 and ZYG-8 interacted by two-hybrid assay, and TAC-1 interacted strongly with ZYG-8ΔKD (*Figure 3—figure supplement 1*). To independently verify these interactions we transfected tagged proteins in HEK293 cells and performed co-immunoprecipitation. We found that TAC-1 co-immunoprecipitated with EFA-6N150, but not with EFA-6N150Δ18 (*Figure 3B*). Likewise, ZYG-8 and EFA-6N150 could be co-immunoprecipitated when coexpressed (*Figure 3C*). These studies suggest that TAC-1 and ZYG-8 specifically interact with the EFA-6 N-terminus. We further tested binding of EFA-6 to TAC-1 and ZYG-8 in cells co-transfected with EFA-6N150, TAC-1 and ZYG-8ΔKD. After immunoprecipitation of EFA-6N150 we could detect both TAC-1 and ZYG-8, and the interactions between EFA-6 and TAC-1 (or ZYG-8) were not affected by the presence of ZYG-8 (or TAC-1) (*Figure 3D*). This result suggests that EFA-6, TAC-1, and ZYG-8 might exist in the same ternary complex.

## TAC-1 and ZYG-8 are required for initiation of axon regrowth after injury and act downstream of EFA-6

TAC-1 and ZYG-8 are essential for embryonic cell division, and *tac-1* and *zyg-8* null mutants are maternal-effect embryonic lethal (*Gönczy et al., 1999*). To examine the roles of these genes in axon regeneration we first used conditional (temperature sensitive, ts) alleles (*Gönczy et al., 2001*; *Bellanger et al., 2007*), here denoted *lf*. When shifted from permissive (15°C) to restrictive (25°C) temperature at L1 stage, *zyg-8(lf)* mutants displayed normal touch axon morphology (not shown). After axotomy in the L4 stage, these animals showed strongly reduced axon regrowth in both PLM and ALM neurons, compared to controls subjected to identical temperature shifts (*Figure 4A*, *Figure 4—figure supplement 1A,B*). *zyg-8(lf)* mutants also showed reduced axon regrowth when maintained at 15°C, even though PLM axon development was normal (*Figure 4—figure supplement 1C,D*), suggesting that axon regrowth is highly sensitive to reduction of *zyg-8* function. The axon regrowth defect in *zyg-8(lf)* mutants was fully rescued by a single copy transgene expressing ZYG-8 driven by a touch neuron specific promoter (*Figure 4—figure supplement 1C,D*), indicating a cell-autonomous function. Similarly, *tac-1(lf)* mutants displayed reduced PLM axon regrowth when shifted from 15 to 25°C in the L1 stage, 20 hr before axotomy (*Figure 4A,B*); *tac-1(lf)* animals raised at 15°C displayed normal regrowth (not shown). As the extent to which these ts mutations impair gene function in post-mitotic cells is not known, we further examined PLM regrowth in *tac-1(ok3305)* null mutants (here denoted *tac-1(0)*) using a genetic mosaic strategy (*Figure 4C*). We rescued the maternal-effect embryonic lethality of *tac-1(0)* mutants with a single copy insertion transgene expressing a floxed version of *tac-1(+)* (*juSi148* or *juSi162*) (see 'Materials and methods'). We then deleted *tac-1(+)* specifically in touch neurons by expressing Cre recombinase under the control of the *mec-7* promoter. Cre-mediated deletion of *tac-1(+)* occurred in 8/8 transgenic (*juSi162; Pmec-7-Cre*) animals (*Figure 4—figure supplement 1E*), suggesting Cre-mediated recombination was efficient. In animals with touch neuron specific deletion of *tac-1*, the PLM axon developed normally, but axon regrowth was impaired to a degree similar to *tac-1(lf)* mutants after L1 upshift (*Figure 4B*), indicating that TAC-1 functions cell autonomously in PLM axon regrowth.

We next tested when ZYG-8 and TAC-1 are required in axon regeneration. We performed temperature upshifts 20 hr before axotomy and examined axon regrowth at 6 hr post axotomy (hpa). *zyg-8(lf)* and *tac-1(lf)* mutants both displayed significantly reduced regrowth at 6 hpa, and significantly fewer regenerative growth cones (defined as axonal tips containing filopodia and/or lamellipodia), compared to controls (*Figure 4D–F*). *tac-1(lf) zyg-8(lf)* double mutants did not show further reduction in axon regrowth, compared to single mutants (*Figure 4A*), suggesting TAC-1 and ZYG-8 could act in a common pathway. *zyg-8(lf) efa-6(0)* double mutants resembled *zyg-8(lf)* single mutants, and *tac-1(lf) efa-6(0)* double mutants resembled *tac-1(lf)* in axon regrowth (*Figure 4A*), consistent with *zyg-8* and *tac-1* acting downstream of *efa-6*. Conversely, overexpression of TAC-1 was sufficient to enhance PLM

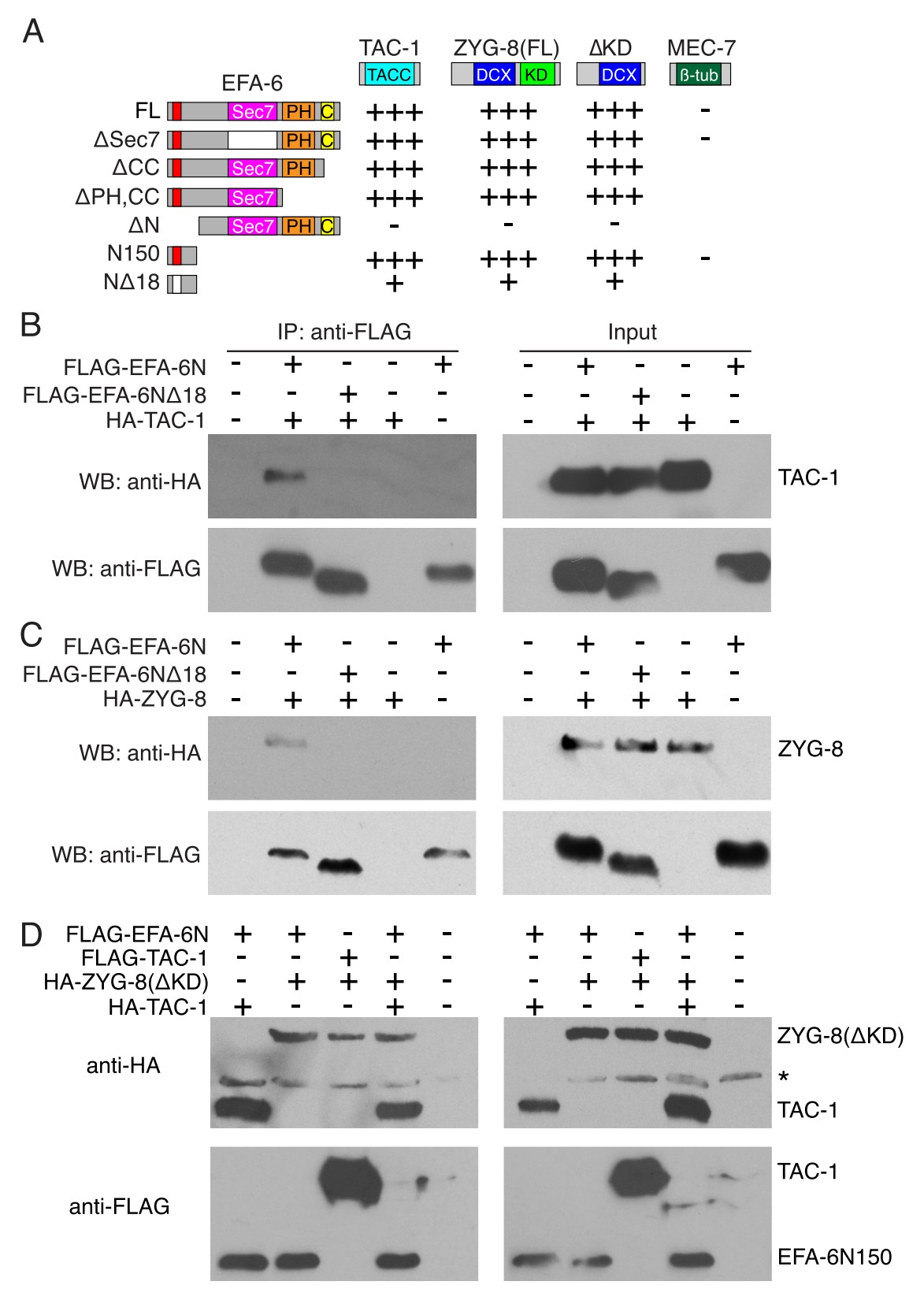

**Figure 3**. EFA-6 interacts with the MT-associated proteins (MAPs) TAC-1 and ZYG-8. (**A**) Summary of two-hybrid analyses. The N-terminus of EFA-6 (N150) is necessary and sufficient for its interaction with TAC-1 and ZYG-8. Deletion of the 18-aa motif from the N-terminus severely impairs binding to TAC-1 and
*Figure 3. continued on next page*

*Figure 3. Continued*

ZYG-8. The interaction between EFA-6 and ZYG-8 does not require the ZYG-8 kinase domain. EFA-6 did not interact with MEC-7/β-tubulin in the two-hybrid assay. '+++', '+', and '−' indicate strong, weak, or undetectable interaction, respectively. (**B**–**D**) Co-immunoprecipitation (Co-IP) of EFA-6 and interactors in HEK293 cells. Indicated constructs were co-transfected into HEK293 cells at a 1:1 ratio. M2-FLAG conjugated magnetic beads were used for IP, and rabbit anti FLAG or anti HA antibodies used for western blotting (WB).

The following figure supplement is available for figure 3:

**Figure supplement 1**. Two-hybrid analysis of the interactions between EFA-6 and TAC-1 or ZYG-8.

regrowth, and did not further enhance the regrowth of *efa-6(0)* mutants (*Figure 4—figure supplement 1F*).

## EFA-6 is required for the injury-induced downregulation of axonal MT dynamics

Our previous analysis showed that loss of EFA-6 function resulted in elevated axonal MT dynamics several hours after injury (*Chen et al., 2011*). To test whether MT dynamics might be influenced by EFA-6 immediately after injury, we examined the acute effects of axon injury on MT dynamics. In wild type animals, axonal EBP-2::GFP comets were dramatically reduced within 50 s post axotomy, consistent with injury triggering rapid MT destabilization (*Figure 5A*). In *efa-6(tm3124)* or *efa-6(ju1200)* mutants we observed slightly increased numbers of EBP-2::GFP comets in uninjured axons, compared to wild type; but these animals did not show a significant reduction in EBP-2::GFP comets immediately after injury (*Figure 5A* and *Figure 5—figure supplement 1A*). The slight reduction in axonal comets after injury in *efa-6(0)* suggests that some MTs can also be downregulated independent of EFA-6. In *efa-6(gf)* animals overexpressing EFA-6N150, the total number of growing MT plus ends in uninjured axons was reduced compared to wild type, and was not further reduced after injury (*Figure 5A*). *tac-1(lf)* mutants had slightly reduced axonal MT dynamics in uninjured axons, and displayed injury-dependent downregulation, whereas *zyg-8(lf)* mutants showed fewer dynamic MTs in uninjured axons, and did not further downregulate MTs after injury (*Figure 5A*). The mild phenotype in *tac-1(lf)* could be due to incomplete loss of function of this ts allele. We further tested the *tac-1(ok3305)* deletion allele using Cre-induced tissue-specific knockout ('Materials and methods'). Compared to control, *tac-1(0)* touch neurons displayed reduced dynamics in uninjured axons similar to *zyg-8(lf)*, and showed no further reduction after injury (*Figure 5—figure supplement 1B*). As reported previously, dynamic axonal MTs are significantly increased at 3 hr post axotomy, and this is further enhanced in *efa-6(0)* mutants (*Chen et al., 2011*) (*Figure 5B*). Neither *tac-1(lf)* nor *zyg-8(lf)* mutants upregulated dynamic axonal MTs by 3 hr post injury (*Figure 5B*). Moreover, MT dynamics in *efa-6(0) tac-1(lf)* or *efa-6(0) zyg-8(lf)* double mutants resembled *tac-1(lf)* or *zyg-8(lf)* single mutants (*Figure 5B*), consistent with EFA-6 functioning upstream of ZYG-8. Thus, axon injury causes an immediate inhibition in growing MTs, dependent on EFA-6 and correlating with its relocalization, followed by a more prolonged increase in growing MTs, dependent on the function of TAC-1 and ZYG-8.

## TAC-1 dynamically responds to injury and colocalizes with EFA-6

We next assessed axonal localization of TAC-1 and ZYG-8 in the steady state and after injury. In wild type uninjured touch neurons, GFP::ZYG-8 was diffuse throughout axon (*Figure 6A*); GFP::TAC-1 or TAC-1::GFP localized to one or two perinuclear spots in the soma and was also diffusely distributed along the axon (*Figure 6B*, *Figure 6—figure supplement 1A*). These patterns were not altered in *efa-6(0)* (*Figure 6A*, *Figure 6—figure supplement 1A*). Conversely, steady state localization of GFP::EFA-6FL or GFP::EFA-6N150 was normal in *tac-1(0)* or *zyg-8(0)* (*Figure 6C*). Thus, in uninjured axons, EFA-6, TAC-1, and ZYG-8 appear to localize independently.

We asked whether TAC-1 and ZYG-8 also responded dynamically to injury. Whereas GFP::ZYG-8 remained diffuse after injury (*Figure 6A*), TAC-1::GFP or GFP::TAC-1 relocalized rapidly to puncta along the axon and in the soma (*Figure 6B*, *Figure 6—figure supplement 1A*). We next co-expressed EFA-6FL::GFP and mKate2::TAC-1 in touch neurons, and found that while EFA-6FL::GFP localized to the plasma membrane throughout the cell, the large perinuclear TAC-1 spots also recruited EFA-6

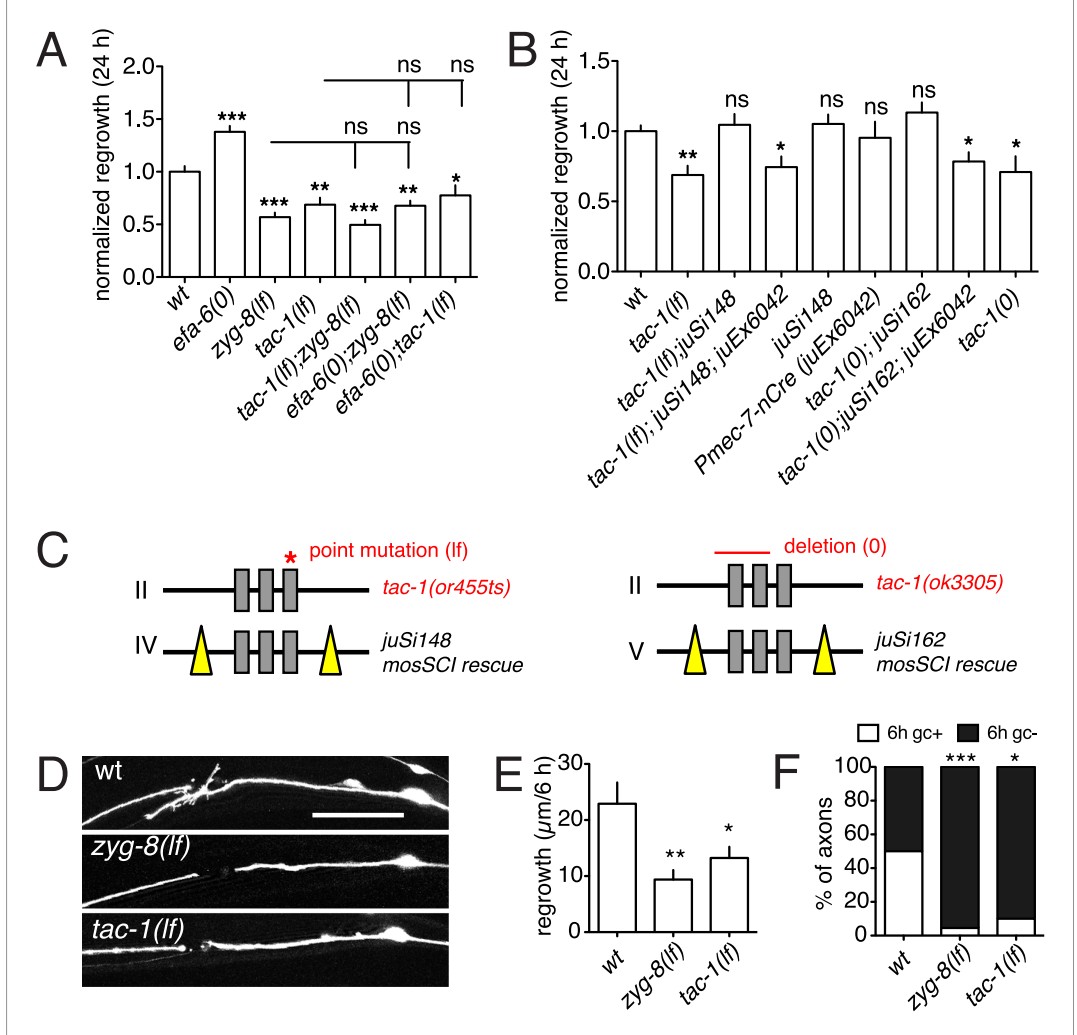

**Figure 4**. TAC-1 and ZYG-8 promote axon regrowth downstream of EFA-6. (**A**) Normalized PLM axon regrowth at 24 hr. Strains were maintained at 15°C, shifted to 25°C 20 hr before axotomy, and kept at 25°C after axotomy for all experiments with ts (lf) alleles. (**B**) Normalized PLM axon regrowth at 24 hr post axotomy. Loss of TAC-1 impairs axon regrowth in a cell-autonomous manner. (**C**) Strategy for neuron-specific deletion of *tac-1* mutants with Mos-SCI single copy transgene of floxed *tac-1*. (**D**) Representative images of axon regrowth at 6 hr post axotomy. WT regrowing axons usually displayed a regenerative growth cone (arrow) at 6 hr post-axotomy whereas *zyg-8(lf)* and *tac-1(lf)* mutant axons rarely display growth cones. (**E**) Quantitation of initial axon regrowth at 6 hr. (**F**) Percentage of axons with regenerative growth cones 6 hr post axotomy. Statistics, one-way ANOVA with Bonferroni post test; ***$p < 0.001$; **$p < 0.01$; *$p < 0.05$; ns, not significant. $n \geq 10$. Scale, 25 μm.

The following figure supplement is available for figure 4:

**Figure supplement 1**. EFA-6 and its interactors regulate axon regeneration.

(*Figure 6D*), suggesting the two proteins can interact in neurons. Localization of EFA-6 to large perinuclear spots was not observed when EFA-6FL::GFP was expressed alone (*Figure 1A*). After injury axonal EFA-6FL::GFP and mKate2::TAC-1 relocalized to largely overlapping puncta (*Figure 6D*); co-expressed EFA-6N150::mKate2 and TAC-1::GFP displayed similar co-localization (*Figure 6—figure supplement 1B*). The injury-induced re-localization of GFP::TAC-1 occurred normally in *efa-6(ju1200)* (*Figure 6B*), as did re-localization of GFP::EFA-6N150 in *tac-1(0)* or *zyg-8(0)* (*Figure 6C*), suggesting that these proteins relocalize independently of each other.

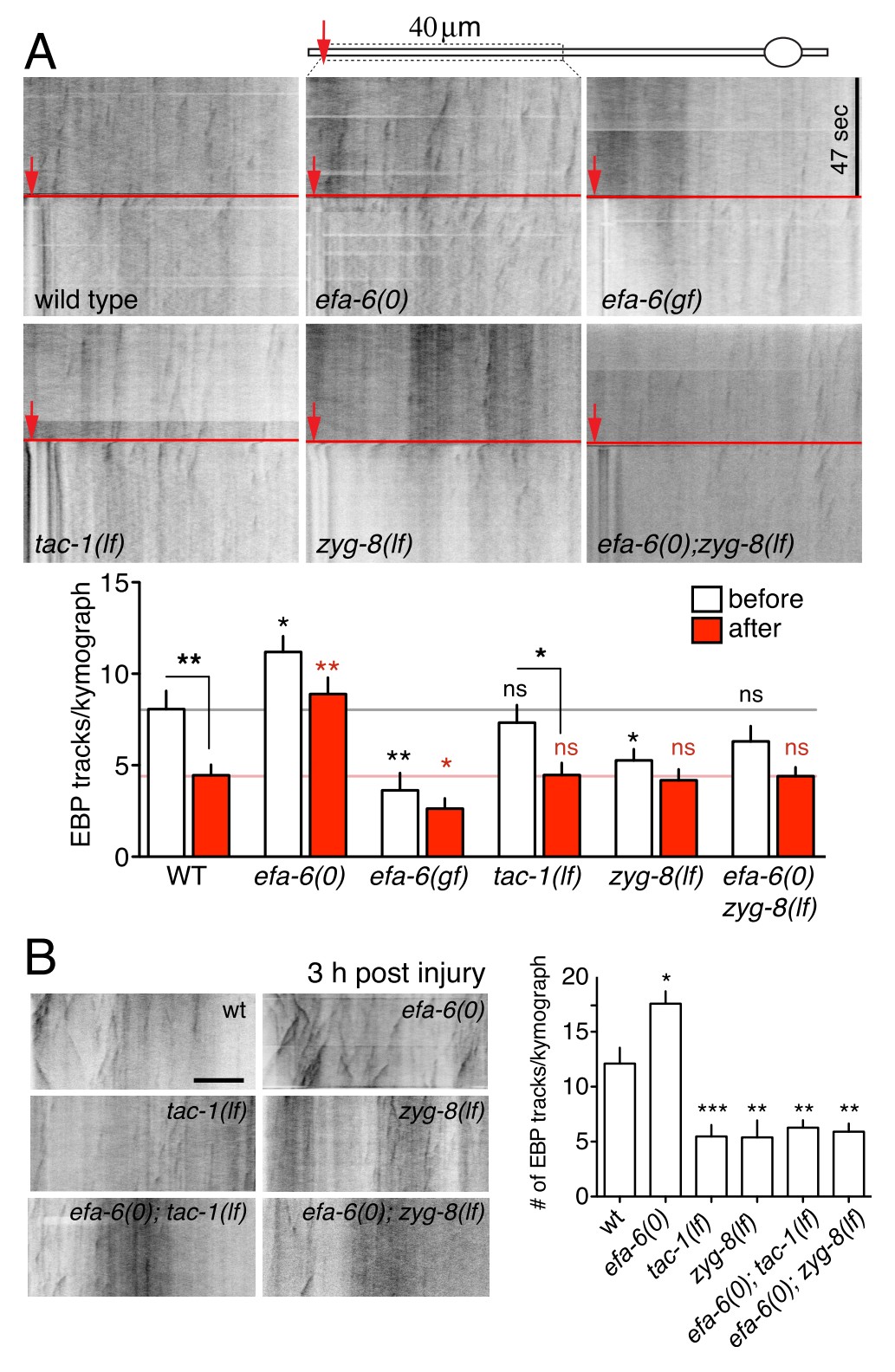

**Figure 5**. Injury triggers rapid down-regulation of MT dynamics dependent on EFA-6. (**A**) MT dynamics (EBP-2::GFP) before and immediately after injury. Kymographs were created from videos of 400 frames (0.23 s/frame), 200 frames before and 200 frames after axotomy. Lower panel: Quantitation of EBP-2::GFP tracks in proximal axon before and immediately after injury. (**B**) MT dynamics 3 hr post injury. Kymographs were created from videos of 200 frames (0.23

*Figure 5. continued on next page*

*Figure 5. Continued*

s/frame). Quantitation of EBP-2::GFP tracks in 40 μm of the proximal axon for 47 s 3 hr post injury. Red line represents time of axotomy; arrow indicates injury site. Strains were maintained at 15°C, shifted to 25°C 20 hr before axotomy. Alleles: *efa-6(tm3124)*, *tac-1(or455*ts*)*, *zyg-8(or484*ts*)*. Statistics, one-way ANOVA with Bonferroni post test; ***$p < 0.001$; **$p < 0.01$; *$p < 0.05$; ns, not significant. $n \geq 10$ axons per condition.

The following figure supplement is available for figure 5:

**Figure supplement 1**. MT dynamics triggered by injury.

## EFA-6 and TAC-1 relocalize to structures containing the MT minus end binding protein Patronin

TAC-1, like other TACC family members, is thought to directly interact with MTs. We considered the possibility that, after injury, TAC-1 and EFA-6 relocalized to axonal MTs. As the punctate localization of TAC-1 and EFA-6 does not resemble that of EBP-2::GFP (i.e., growing MT plus ends), we tested whether TAC-1 and EFA-6 were becoming localized to MT minus ends. PTRN-1 is the *C. elegans* member of the Patronin/CAMSAP family, known to bind to and stabilize MT minus ends (*Meng et al., 2008*; *Goodwin and Vale, 2010*; *Yau et al., 2014*). In *C. elegans* neurons PTRN-1 localizes to axonal puncta, likely the sites of MT minus ends (*Chuang et al., 2014*; *Marcette et al., 2014*; *Richardson et al., 2014*). GFP::PTRN-1 localization does not dramatically change after axon injury (*Chuang et al., 2014*), while in the same time period EFA-6N150::mKate2 became punctate and partially colocalized with GFP::PTRN-1, independent of the tagged reporters (*Figure 7A*, *Figure 7—figure supplement 1B*). Similarly, after injury TAC-1::mKate2 became highly colocalized with GFP::PTRN-1 (*Figure 7—figure supplement 2*). These observations suggest that axon injury causes TAC-1 to relocalize to PTRN-1-containing puncta, and causes EFA-6 to relocalize to regions overlapping with or closely adjacent to the TAC-1/PTRN-1 puncta.

We further asked whether injury-induced relocalization of EFA-6 or TAC-1 required PTRN-1. In a *ptrn-1* null mutant GFP::EFA-6N150 or TAC-1::GFP relocalized after injury as in wild type (*Figure 7—figure supplement 1C,E*). GFP::EFA-6N150 localization in the *tac-1(0) ptrn-1(0)* double mutant was normal either before or after injury (*Figure 7—figure supplement 1D*). Thus, although TAC-1 and EFA-6 relocalize to PTRN-1-containing puncta and adjacent regions respectively, their recruitment does not absolutely require PTRN-1. The relocalization of EFA-6 and TAC-1 may involve multiple redundant factors.

*ptrn-1* null mutants display largely normal PLM outgrowth and significantly reduced axon regeneration (*Chuang et al., 2014*). We found that *efa-6(0) ptrn-1(0)* double mutants resembled *ptrn-1(0)* single mutants in regeneration (*Figure 7B*); conversely, *tac-1(lf) ptrn-1(0)* double mutants were not further enhanced, compared to either single mutant (*Figure 7B*). These double mutant analyses are consistent with PTRN-1 and TAC-1 acting in a common pathway in regeneration, with EFA-6 acting as a negative regulator of one or both proteins.

## Discussion

We previously identified EFA-6 as a potent intrinsic inhibitor of axon regeneration (*Chen et al., 2011*). Here we dissected how EFA-6 regulates MT dynamics in the axonal response to injury, yielding the following insights: (1) injury triggers a rapid relocalization of EFA-6 from the cell membrane to subcellular structures overlapping with the MT minus end binding protein PTRN-1/Patronin; (2) injury also triggers a rapid local downregulation of axonal MT dynamics, a process that requires EFA-6 and which occurs simultaneously with EFA-6 relocalization; (3) a conserved motif in the otherwise unstructured EFA-6 N-terminus is essential for its relocalization and for its effects on axonal MT dynamics; (4) the EFA-6 N-terminus directly interacts with two highly conserved MAPs, TAC-1/TACC and ZYG-8/DCLK, both of which are required for normal regrowth and for the later phase of upregulated axonal MT dynamics. Our studies reveal that axonal injury triggers an intricate sequence of alterations in axonal MT dynamics. Precise control of the axonal MT response by positive and negative regulators of MT dynamics appears critical for the conversion of the mature axon into a regrowing growth cone.

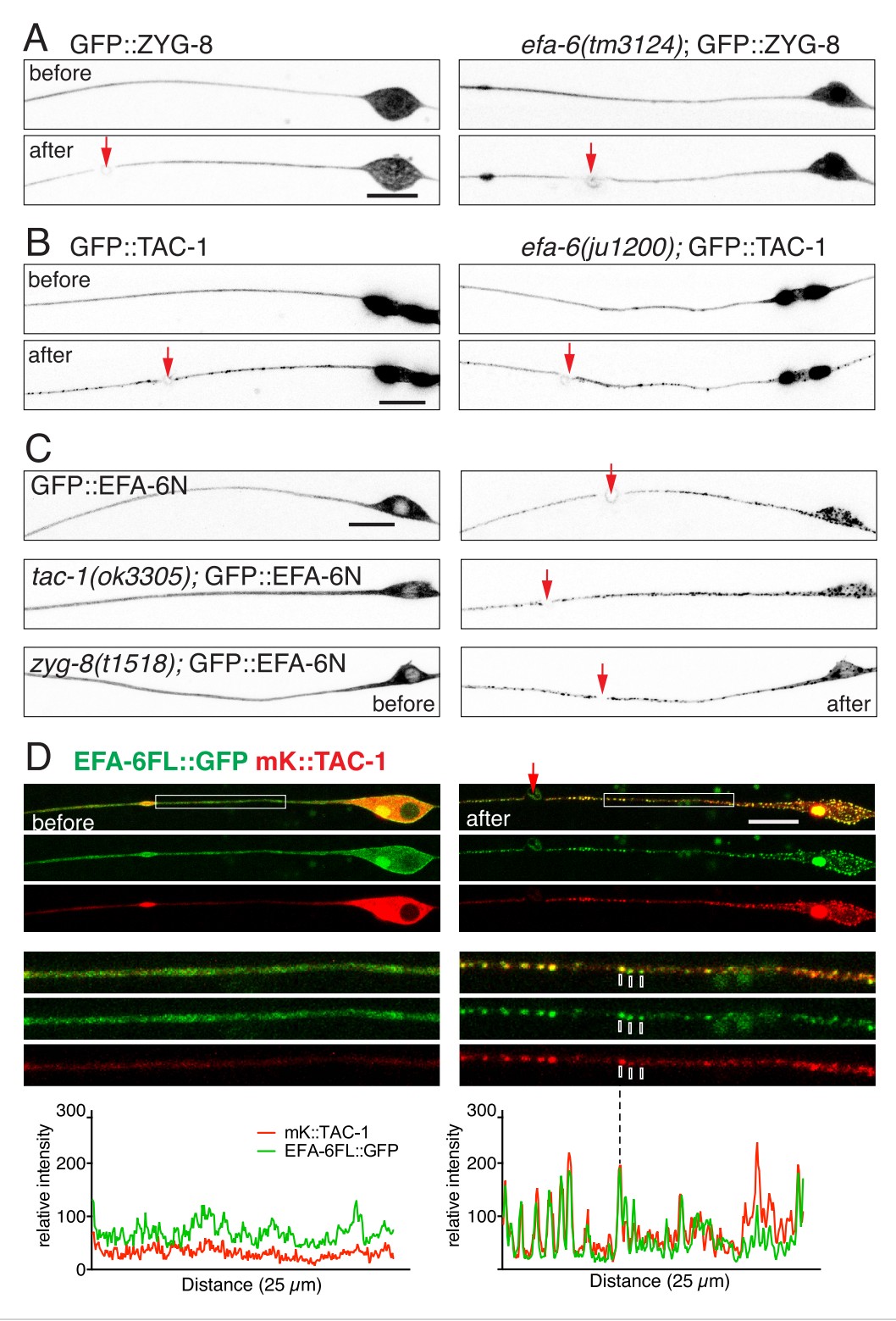

**Figure 6**. TAC-1 relocalizes in response to injury to become co-localized with EFA-6. (**A**) Localization of GFP::ZYG-8 in touch neurons before and 2 min after axotomy in wild type and *efa-6(tm3124)*. GFP::ZYG-8 localization is not affected by axon injury or loss of EFA-6. Transgene: P*mec-4*-GFP::ZYG-8*(juEx5932)*. (**B**) GFP::TAC-1 in PLM before and 2 min after axotomy at wild type and *efa-6(ju1200)* backgrounds. Injury triggered relocalization of TAC-1 was similar to EFA-6 and not dependent on EFA-6. P*mec-4*-GFP::TAC-1*(juEx5759)*. (**C**) GFP::EFA-6N150 (*juEx3531*)
*Figure 6. continued on next page*

*Figure 6. Continued*

localization in wild-type, *tac-1(ok3305)* and a putative *zyg-8* null allele *zyg-8(t1518)* (*Gönczy et al., 2001*). Relocalization of EFA-6N150 was not dependent on TAC-1 or ZYG-8. (**D**) Localization of EFA-6FL::GFP and mKate2::TAC-1 before and after axotomy in a touch neuron. Before axotomy, TAC-1 was diffuse in soma and along the axon, and concentrated in a large perinuclear dot. EFA-6 was predominantly localized to the plasma membrane and also in the perinuclear dot marked by TAC-1. After axotomy, both proteins became punctate and the puncta were partially co-localized; enlargements in small boxes below. Graphs of line scans along the axon are shown below the enlarged images. Arrow, injury site; scale, 10 μm.

The following figure supplement is available for figure 6:

**Figure supplement 1**. Injury triggered relocalization.

## EFA-6 functions in the steady state and injured axon

EFA-6 induces catastrophe or pausing of growth at MT plus ends at the cortex of embryonic cells (*O'Rourke et al., 2010*), and our analysis is consistent with this model in steady-state (uninjured) axons. In mature neurons EFA-6 localizes to the cell membrane via its PH domain. *efa-6((0))* mutant axons display elevated numbers of growing MTs in the steady state, and display impenetrant developmental overgrowth, indicating that in the absence of injury EFA-6 restrains axonal MT dynamics and mildly inhibits axon outgrowth. These steady-state roles of EFA-6 are mediated by the N-terminus, as they are fully rescued by expression of the N-terminal 150 aa. Overexpression of either full length EFA-6 or the N-terminus causes axons to undershoot, whereas overexpression of EFA-6ΔN causes axons to overshoot. The opposing effects of overexpression of the N-terminus and the C-terminus suggest that in the steady state the EFA-6 N-terminus might be inhibited by the remainder of the protein.

After axon injury, EFA-6 displays a dramatic and specific transient relocalization to punctate structures associated with MT minus ends. Interestingly, this relocalization does not require membrane association, as EFA-6 fragments lacking the PH domain, or containing only the N-terminal 1–70 aa, were not membrane-localized, yet became relocalized to axonal puncta after injury. As relocalized full-length EFA-6 appears to remain membrane associated, it is possible that after injury EFA-6 localizes to the cytoskeleton via its N terminus while remaining membrane associated via its PH domain. Speculatively, injury signals may increase EFA-6 N-terminus activity by releasing it from inhibition by the EFA-6 C-terminus.

## The EFA-6 N-terminus is an intrinsically disordered region

The N-termini of EFA6 family members display low sequence complexity and minimal primary sequence similarity, with the exception of the 18 aa motif found in invertebrate family members (*O'Rourke et al., 2010*). The N-termini of *C. elegans*, *Drosophila* and mammalian EFA6 proteins all have a high probability of intrinsic disorder. Intrinsically disordered proteins (IDPs) and disordered protein regions are increasingly recognized as having important biological activities (*Oldfield and Dunker, 2014*). Well-studied examples of IDPs in the nervous system include the MT-binding proteins Tau (*Schweers et al., 1994*) and stathmin (*Honnappa et al., 2006*). Intrinsic disorder does not imply a lack of structure, but rather allows these regions to function as binding surfaces with multiple interacting partners; IDPs are often hubs in protein–protein interaction networks (*Cumberworth et al., 2013*). In the EFA-6 N-terminus the 18 aa motif is predicted to have relative structural order, and might act as a molecular recognition feature as in other IDPs. The 18 aa motif is critical for EFA-6 relocalization and for the EFA-6 N-terminus to interact with ZYG-8 and TAC-1. Moreover, as mutation of two or more residues within the 18 aa motif impairs its activity, both the exact sequence of the 18 aa motif and the surrounding extended intrinsically disordered domain appear to be critical for the ability of the EFA-6 N-terminus to regulate MT dynamics. In some assays, the requirement for the 18 aa motif in EFA-6 function was not all-or-none, suggesting it may facilitate the function of the surrounding interacting domain. Moreover, neither ZYG-8 nor TAC-1 may directly bind the 18 aa motif; the motif may be important for correct folding of the larger N-terminal interacting domain.

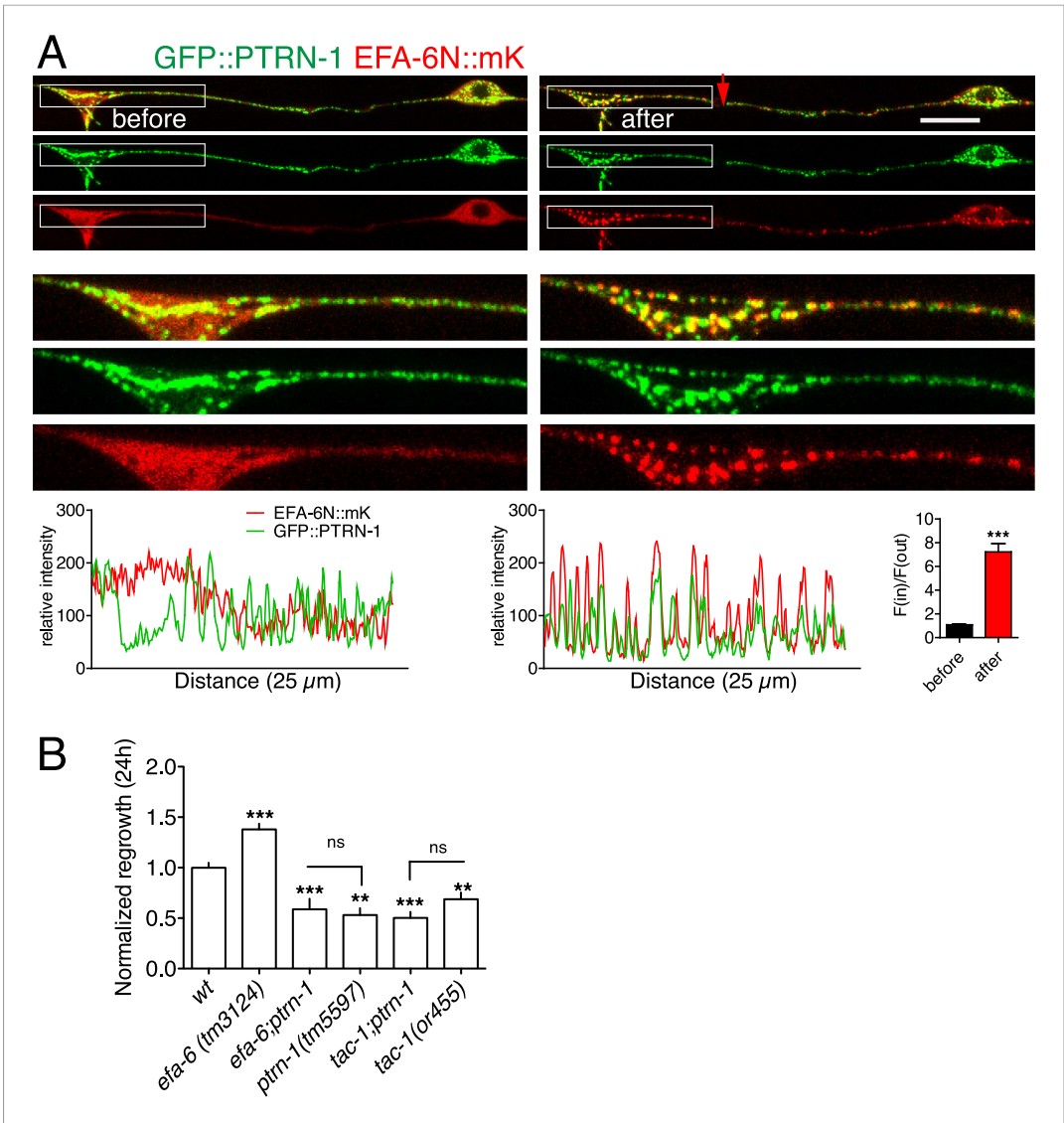

**Figure 7**. EFA-6 and TAC-1 re-localize to puncta overlapping with the MT minus end-binding protein Patronin/PTRN-1.
(**A**) Localization of PTRN-1 and EFA-6N150 in PLM before and after axotomy. EFA-6N150::mKate2 was diffuse in soma
and axon before injury, and became punctate after injury and these puncta co-localized to GFP::PTRN-1. Enlarged
images of the regions in boxes are shown below. Graphs of line scans along the axon and F(in)/F(out) ratio quantitation
are shown below. Increased F(in)/F(out) ratio indicates higher degree of colocalization post axotomy; see
*Figure 7—figure supplement 1A* and 'Materials and methods' for calculation of F(in)/F(out). Statistics: Student's *t*-test.
***p < 0.001. (**B**) Epistatic interactions between *efa-6(0)*, *ptrn-1(0)*, and *tac-1(lf)*. Normalized PLM regrowth. Strains
without or with temperature shift (cultured at 15°C and upshifted to 25°C 20 hr before axotomy and kept at 25°C for 24
hr after axotomy) were quantified separately. Statistics: one-way ANOVA with Bonferroni post test.

The following figure supplements are available for figure 7:

**Figure supplement 1**. Co-localization of EFA-6 with PTRN-1 after injury.

**Figure supplement 2**. TAC-1 relocalizes to PTRN-1 puncta after injury.

## The EFA-6 N-terminus binds MAPs ZYG-8/DCLK and TAC-1/TACC

Our analysis suggests the EFA-6 N-terminus regulates MT dynamics indirectly via the MAPs ZYG-8
and TAC-1. Identification of TAC-1 and ZYG-8 as EFA-6 interactors was unexpected, as in embryonic

cells TAC-1 is predominantly centrosomal or cytoplasmic (*Le Bot et al., 2003*), and ZYG-8 localizes along MTs (*Bellanger et al., 2007*), whereas EFA-6 is cortically localized (*O'Rourke et al., 2010*). TAC-1 and ZYG-8 interact physically, but localize independently (*Bellanger et al., 2007*). However EFA-6, TAC-1, and ZYG-8 are all present in axons, suggesting these proteins may interact directly in differentiated cells. TAC-1 and EFA-6 partly colocalize after injury, consistent with their interaction being regulated by injury. Although ZYG-8 localization appeared unaffected by injury, it is predicted to associate along the length of MTs, and so could also interact with EFA-6 after injury. Notably, loss of function of TAC-1 or ZYG-8 did not detectably affect developmental axon outgrowth, but strongly blocked axon regeneration, indicating regenerative regrowth is highly dependent on these MAPs.

Neither TACCs nor ZYG-8/DCLK exclusively associate with MT plus ends. TACC proteins can act both at plus and minus ends of centrosomal MTs (*Lee et al., 2001*; *Peset and Vernos, 2008*). Localization of human TACC3 to minus ends is regulated by Aurora A dependent phosphorylation (*Barros et al., 2005*; *LeRoy et al., 2007*). Conversely, Doublecortin (DCX) decorates the length of the MT lattice and stabilizes it (*Moores et al., 2006*). DCX can also track MT plus ends, and acts as a nucleation factor, stabilizing polymerization intermediates (*Moores et al., 2004*; *Bechstedt and Brouhard, 2012*). Like DCX, DCLK is thought to be able to interact with MTs along their length (*Burgess and Reiner, 2000*). Our results suggest that after axon injury, TAC-1, EFA-6, and possibly ZYG-8 may interact with one another at specific subregions of MTs at or adjacent to minus ends. As EFA-6 has opposite effects to TAC-1 and ZYG-8, EFA-6 may transiently inhibit the activity of TAC-1 or ZYG-8, resulting in the rapid disruption of axonal MT growth. Following this initial phase, EFA-6 returns to its steady state, relieving the inhibition of TAC-1 and ZYG-8, which are then required for the later upregulation of axonal MT dynamics.

## EFA-6 and TAC-1 relocalize to sites of MT minus ends

CAMSAPs/Patronins directly bind MT minus ends and protect them from depolymerization by Kinesin-13 (*Tournebize et al., 2000*; *Meng et al., 2008*; *Goodwin and Vale, 2010*; *Hendershott and Vale, 2014*; *Jiang et al., 2014*). The minus end protection activity of CAMSAP family proteins is important for their function in maintaining noncentrosomal MTs. In *C. elegans* axons PTRN-1 localizes to puncta that are thought to define sites of MT minus end anchoring or stabilization (*Marcette et al., 2014*; *Richardson et al., 2014*). Strikingly, EFA-6 and TAC-1 relocalize close to or at these sites after injury, suggesting MT minus ends may be an important site of regulation. As PTRN-1 itself is not required for EFA-6 or TAC-1 relocalization, other proteins may be involved in their targeting. Indeed, the near-normal development and behavior of *ptrn-1* null mutants suggests additional factors can stabilize MT minus ends in noncentrosomal arrays. Like TAC-1 and ZYG-8, PTRN-1 is required for axon regrowth. EFA-6 or its interactors might modulate the function of PTRN-1 in axon regrowth.

## EFA-6 and injury-triggered MT dynamics

MT dynamic instability, first studied as a function of tubulin concentration in vitro, is influenced by a wide array of positive and negative regulators in vivo. For example, in *Xenopus* egg extracts, MT dynamics are determined by a balance between the MT growth-promoting XMAP215 and MT-destabilizing XKCMI (*Tournebize et al., 2000*; *Kinoshita et al., 2001*). Our findings suggest that the initial stages in axon regeneration are also driven by a sequence of shifts in the balance between opposing activities of MT destabilizers such as EFA-6 and MT growth promoting factors such as TAC-1 and ZYG-8.

Axonal injury in *C. elegans* triggers a highly regulated sequence of changes in MT dynamics that correlate closely with changes in EFA-6 localization and activity. We did not detect significant down-regulation of MT growth in response to injury in the absence of EFA-6, using two independent alleles. Although a small decrease in MT dynamics was observed in *efa-6* mutants, this was not statistically significant, and calculations of statistical power suggest that changes of >15% should be detectable in experiments of the sample size used here. Nevertheless there may be EFA-6-dependent and EFA-6-independent effects on MT dynamics immediately after injury. Changes in MT dynamics within seconds of axon injury have been studied in *Aplysia* neurons (*Erez et al., 2007*; *Erez and Spira, 2008*), which display rapid local MT depolymerization followed by repolymerization over several minutes. *Drosophila* neurons also display acute and chronic alterations in MT dynamics after injury (*Chen et al., 2012*; *Lu et al., 2015*). Recent in vivo imaging of mammalian axons found an acute increase in axonal MT dynamics after laser axotomy, followed by a sustained increase over

several days (*Kleele et al., 2014*). Thus, the exact sequence of MT dynamics changes after injury may vary between cell types and organisms. An important future goal will be to address the role of EFA6 family members in mammalian axon regeneration, and whether manipulation of this MT regulatory pathway can enhance regeneration in therapeutic settings.

## Materials and methods

### *C. elegans* genetics

We maintained *C. elegans* following standard methods. Transgenes were introduced into mutant backgrounds by crossing or injection; homozygosity for all mutations was confirmed by PCR or sequencing. We used the following published transgenes: P*mec-7*-GFP(*muIs32*), P*mec-4*-GFP(*zdIs5*), P*mec-4*-EBP-2::GFP(*juIs338*) (*Chuang et al., 2014*).

### Molecular biology and transgenes

We made new plasmids by Gateway recombination (Invitrogen / Life Technologies, Grand Island, NY) or Gibson assembly, as listed in *Supplementary file 1A*; new transgenes are listed in *Supplementary file 1C*. We amplified cDNAs from existing clones or from total first-strand cDNA; all clones were sequenced. Mutations were introduced by Quikchange mutagenesis (Agilent Technologies, Santa Clara, CA). We followed standard procedures to generate multicopy extrachromosomal transgenes; plasmids were injected at 1–30 ng/µl, and co-injection markers at 75 ng/µl. We analyzed 3 to 5 lines per construct. We made single copy insertions using Mos-SCI (http://www.wormbuilder.org/), on chromosomes IV (strain EG8081) or V (EG8083).

### Live imaging, laser axotomy, and FRAP

We collected fluorescence images on Zeiss LSM710 or LSM510 confocal microscopes. We performed laser axotomy as described (*Chen et al., 2011*). We performed live imaging and analysis of EBP-2::GFP dynamics as described (*Ghosh-Roy et al., 2012*); in some experiments we immobilized animals in 30 mM muscimol on pads of 10% agarose in M9.

For quantitative analysis of protein localization and colocalization, animals were immobilized with either 0.7% phenoxypropanol or 30 mM muscimol. Confocal z-stacks were collected with 0.5 µm intervals. Typically 3–4 slices span an axon (1–2 µm diameter). Images were analyzed using Zeiss Zen and Metamorph (Molecular Devices, Sunnyvale, CA). Briefly, we drew lines along the axon, starting at the injury site or soma, then used the line scan tool to measure the average fluorescence intensity of 8 pixels (∼0.7 µm) surrounding the lines. For puncta number (or peak #) analysis, we counted any peak with intensity greater than the mean +1 SD as a punctum. To quantitate colocalization, we measured the F(in)/F(out) ratio (see *Figure 7—figure supplement 1A*) using Metamorph software. A small ROI (region of interest) was drawn to cover one punctum of GFP::PTRN-1. Average intensity of mKate2 (EFA-6N150) within the ROI was measured as F(in). The ROI was then duplicated to cover a small region in the axon with no GFP::PTRN-1 puncta, and average intensity of mKate2 (EFA-6N150) within this ROI was measured as F(out). F(in)/F(out) was then calculated as [F(in) − background intensity]/[F(out) − background intensity]. Before injury, EFA-6N::mKate2 is evenly distributed in the axon, and mKate2 intensity inside and outside of the GFP::PTRN-1 puncta is similar in level, so the F(in)/F(out) ratio is close to 1. 2 min post injury, EFA-6N is relocalized to PTRN-1 puncta, so mKate2 intensity within the PTRN-1 puncta is much higher than outside the puncta, resulting in a significantly higher F(in)/F(out) ratio.

For FRAP we set circular regions of interest (ROIs) for acquisition and photobleaching, using 2% laser power for acquisition and 100% laser power (488 nm, Zeiss LSM710) for photobleaching. We acquired 5 and 25 images before and after photobleaching. We chose ROIs 1 µm diameter from regions of median initial intensity in the soma or axon. In videos of uninjured neurons, we placed ROIs in regions with diffuse or relatively enriched GFP signal; in injured neurons, ROIs were drawn around puncta. Average fluorescence intensity in each ROI at each frame was measured in Zen. To generate FRAP curves we normalized intensity to the average of the five frames prior to photobleaching. We calculated $t_{1/2}$ following standard formulas; the immobile fraction was calculated by the Zen program.

## Yeast two-hybrid screening and assays

We performed two-hybrid screening as described (*Wang et al., 2013*). We cloned EFA-6 full-length cDNA and fragments into a pMB27-Gal4-BD-gtwy vector, derived from the pPC97-Gal4-BD vector. We transformed baits into yeast strain Y8930, and mated these to a pPC86-Gal4-AD prey library of mixed-stage *C. elegans* cDNAs in strain Y8800. Plasmids for two-hybrid experiments are listed in *Supplementary file 1B*. We screened >2 × 10$^6$ independent colonies per bait, and identified interacting cDNAs by plasmid amplification and sequencing. To test specific interactions we cloned the appropriate full length or fragment cDNAs into the pACT2 (Gal4 activation domain) or pBTM116 (LexA DNA-binding domain) vectors (Clontech, Mountain View, CA) and co-transformed constructs into yeast strain L40. We grew transformed yeasts on agar plates with SD medium (synthetic minimal medium) lacking leucine and tryptophan; interactions were examined on plates with SD medium lacking leucine, tryptophan, and histidine, with or without 3-AT.

## Coimmunoprecipitation in HEK293 cells

Plasmids used in co-immunoprecipitation experiments are listed in *Supplementary file 1B*. We co-transfected FLAG-tagged EFA-6N150 or EFA-6N150Δ18, and HA-tagged TAC-1 or ZYG-8 into HEK293 cells using X-tremeGene 9 DNA Transfection Reagent (Roche Diagnostics Corporation, Indianapolis, IN). 48 hr after transfection, cells were lysed using lysis buffer (25 mM Tris-HCl pH 7.4, 150 mM NaCl, 1 mM EDTA, 1% NP-40 and 5% glycerol). Anti-FLAG M2 antibody conjugated magnetic beads (Sigma M8823, Sigma-Aldrich, St Louis, MO) were used for IP; anti-HA (rabbit) (Abcam ab9110, Abcam, Cambridge, UK) and anti-FLAG (rabbit) (Sigma F7425) were used for western blotting.

## CRISPR targeted deletion

We used CRISPR based gene targeting (*Dickinson et al., 2013*) to delete the genomic region encoding the EFA-6 18 aa motif. Briefly, we obtained Cas9-sgRNA plasmid from the Goldstein lab and inserted an sgRNA sequence targeting *efa-6* into the vector using Quikchange mutagenesis. The two sgRNA sequences were GGCGAGGGGCTCCATCAATGG and GATGCAACTGTGGTACCTGG, targeting *efa-6* exon 1 and exon 2 respectively. 50 ng/µl of each Cas9-sgRNA plasmid and 20 ng/µl P*sur-5*-mCherry were co-injected into wild type animals. From 15 F$_1$ progeny we found one animal heterozygous for *efa-6(ju1200)*, which deletes 500 bp of exon 1, intron 1, and exon 2, and has a 26 bp insertion. mRNAs produced in *efa-6(ju1200)* encode polypeptides with a premature stop codon after amino acid 15, eliminating the 18 aa motif and the rest of EFA-6.

## Locomotion analysis

We measured locomotion velocity using WormTracker 2.0 (*Brown et al., 2013*) NGM plates were seeded with OP50 bacteria 3 hr before experiments. Individual young adult worms were picked gently from the culture plate to a fresh tracking plate. 1 min later, the plate was placed on the worm tracker platform and locomotion recorded for 1 min at 10 frames per second for each animal.

## Statistical analysis

We used Prism (GraphPad Software, La Jolla, CA) for all statistical analysis. A two-tailed Student's *t*-test was used for comparisons of two groups. For multiple comparisons we used one-way ANOVA with Bonferroni post test. To compare variables such as growth cone percentage we used the Fisher exact test.

## Acknowledgements

We thank members of the Jin and Chisholm labs for discussion and advice. We thank Bob Goldstein for the Cas9 plasmid. We thank Miriam Goodman for communicating unpublished results on *tac-1*, and Bruce Bowerman for discussions. LC was supported by the Neuroplasticity of Aging Training Grant NIH T32 AG000216. MC was supported by the UCSD Cellular and Molecular Genetics Training Grant (NIH T32 GM007240). MB and TK were supported by NWO/ALW Vidi grant 864.09.008. LC is an Associate and YJ is an Investigator of the Howard Hughes Medical Institute. Supported by NIH R01NS093588, R01NS057317 and R56NS057317 to ADC and YJ.

# Additional information

## Funding

| Funder | Grant reference | Author |
|---|---|---|
| National Institutes of Health | R56 NS057317, R01 NS057317 | Yishi Jin, Andrew D Chisholm |
| Howard Hughes Medical Institute | | Lizhen Chen, Yishi Jin |
| NWO/ALW Vidi | 864.09.008 | Thijs Koorman, Mike Boxem |
| National Institutes of Health | R01 NS093588 | Yishi Jin, Andrew D Chisholm |
| National Institutes of Health | Neuroplasticity of Aging Training Grant NIH T32 AG000216 | Lizhen Chen |
| National Institutes of Health | UCSD Cellular and Molecular Genetics Training Grant NIH T32 GM007240 | Marian Chuang |

The funders had no role in study design, data collection and interpretation, or the decision to submit the work for publication. Should this section also include the training grant info listed in Acknowledgments?

## Author contributions

LC, Conception and design, Acquisition of data, Analysis and interpretation of data, Drafting or revising the article; MC, TK, Acquisition of data, Analysis and interpretation of data; MB, Acquisition of data, Analysis and interpretation of data, Drafting or revising the article; YJ, ADC, Conception and design, Analysis and interpretation of data, Drafting or revising the article

## Author ORCIDs

Mike Boxem, http://orcid.org/0000-0003-3966-4173
Andrew D Chisholm, http://orcid.org/0000-0001-5091-0537

# Additional files

## Supplementary file

• Supplementary file 1. (**A**) Plasmids for *C. elegans* Transgenes. (**B**) Plasmids for Yeast Two-Hybrid and Co-immunoprecipitation. (**C**) *C. elegans* strains, transgenes, and clones.

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
