## [Decision Letter]

Thank you for submitting your work entitled “Axon injury triggers EFA-6 mediated destabilization of axonal microtubules via TACC and doublecortin like kinase” for peer review at *eLife*. Your submission has been favorably evaluated by K VijayRaghavan (Senior Editor) and two reviewers.

The reviewers have discussed the reviews with one another and the editor has drafted this decision to help you prepare a revised submission.

An important and compelling topic in the field of neuroscience and regenerative medicine is the molecular and cellular mechanisms that inhibit axons from regrowth following traumatic injury. This is a massive and detailed manuscript that makes a significant contribution to our understanding of neuronal cell biology following axon injury. Chen et al. examine this topic in the model *C. elegans* using a combination of genetics, cell biology, and biochemistry. In a previous publication, the authors, through a genetic screen, identified a role for the Arf GEF EFA-6 in inhibiting the regrowth of *C. elegans* axons following injury by laser axotomy. EFA-6 had previously been shown by the Bowerman lab to inhibit microtubule growth at the cortex of *C. elegans* embryos through a novel domain in the N-terminus of the protein; however, neither the paper from the Bowerman lab nor from the previous authors had addressed the underlying molecular mechanism. This current manuscript proposes such a molecular mechanism.

The authors present several lines of evidence that axonal injury triggers the rapid subcellular localization of EFA-6 to microtubule (MT) minus ends. They show that EFA-6 inhibits the number of growing MT plus ends along axons under basal conditions, and that injury can further depress the number of growing MT plus ends via a process that requires, in part, EFA-6. They show that the N-terminal motif described initially in the Bowerman paper binds directly to two conserved MT-associated proteins: TAC-1 (TACC) and ZYG-8 (DCLK). They find that TAC-1 and ZYG-8 are both required for axon regrowth following injury, as well as for the overgrowth phenotype observed when *efa-6* is knocked out. In addition, they show that ZYG-8 is required for EFA-6 to regulate the number of growing MT plus ends. They also show that injury promotes the colocalization of TAC-1 with EFA-6 to MT minus ends in response to injury, although it appears that the two proteins do not regulate each other's subcellular localization. The authors do not show that the subcellular localization of either EFA-6 or TAC-1 to MT minus ends is required for axon regrowth; however, they do show that the Patronin homolog PTRN-1, which normally stabilizes MT minus ends, is required for regrowth, so a role for EFA-6 and/or TAC-1 at the MT minus ends certainly remains a possibility. Taken together, the authors' data suggest that injury triggers acute MT plus end catastrophe through the inhibition of ZYG-8 by EFA-6, followed by a slower MT plus end rescue and regeneration that is dependent on both ZYG-8 and TAC-1, and that these MT dynamics are required for proper axonal regrowth.

Overall, the manuscript is well written and interesting, and it clearly moves beyond the previous publications on EFA-6 to get at molecular and cell biological mechanism. A published version of this manuscript would be of high interest to those in the fields on axon regeneration and MT dynamics in neurons. That said, the manuscript does require a few alterations before publication. We outline substantive comments below.

1) A key negative result is the lack of significant change in EBP tracks after axon injury in *efa-6* mutants (Figure 5). Given the importance of this negative result to the model, the power of the experiment should be discussed. Ideally it would be nice to see some independent confirmation, perhaps with another allele.

2) Figure 7—figure supplement 1, part A has a cartoon illustrating quantification of co-localization. However, there is no formal description defining what exactly is being measured by F^in^ and F^out^, or how the F^in^/^out^ ratio is calculated. The Results section refers to the Methods, but the Methods section does not really describe the details. The authors should specify the details of how this ratio is calculated.

3) The authors perform a really nice touch neuron-specific knockout experiment for TAC-1 using a single-copy, MOSCI-based, floxed, wild-type TAC-1 transgene (called *juSi162*) and a transgene that expresses Cre recombinase solely in the touch neurons. They have a PCR strategy for verifying that the recombination occurs as described (found in Figure 4—figure supplement 1). However, the specifics of the PCR assay and the primers are not described in the paper. It's unclear if the worms from panel E also contain the *tac-1(ok3305)* deletion mutation. Presumably they do, and that the middle black primer falls within the deletion such that no PCR products are from the endogenous gene? The authors should state this and describe the experiment in more detail.

4) Figure 6 is difficult for the reader to navigate back and forth from the main figure and the supplement. More importantly, there are interesting and important results in the supplement that really should be in the main Figure (i.e., that injury does not alter ZYG-8 subcellular localization, and that TAC-1 and EFA-6 subcellular localization are not dependent on one another). The authors should move the panels A, B, and E from Figure 6—figure supplement 1 into the main body of Figure 6.

5) Figure 5. The authors do not observe a change in MT plus end growth in *tac-1(lf)* mutants in the graph under panel **A**. Presumably this experiment was done with the *ts* allele? What happens in the deletion allele examined with the Cre recombinase approach? This later genotype is more likely to be a null. This experiment could address whether or not TAC-1 is required for the EFA-6-mediated changes in MT dynamics (similar to what is observed for ZYG-8). Also, for the graph in panel B, why don't the authors show the data for the *efa-6(lf)*; *zyg-8(lf)* and *efa-6(lf)*; *tac-1(lf)* double mutants?

6) The authors identified ZYG-8 and TAC-1 as EFA-6 binding partners through a yeast two-hybrid screen. They then confirmed the interaction using GST-pull downs of proteins expressed in HEK293 cells. If this were a study being conducted in mammalian brain, one might suggest that they verify whether such protein complexes can be isolated between endogenous proteins from intact brain. However, that would be an unreasonable request given that the *C. elegans* nervous system comprises such a small percentage of the total biomass of the animal and there is no way to dissect out large quantities of worm neurons for such biochemical studies. Such an experimental requirement should not be requested of these authors, and the significance of their in vivo genetic findings far outweigh the absence of data supporting protein–protein interactions of the endogenous proteins.

---

## [Author Response]

*Overall, the manuscript is well written and interesting, and it clearly moves beyond the previous publications on EFA-6 to get at molecular and cell biological mechanism. A published version of this manuscript would be of high interest to those in the fields on axon regeneration and MT dynamics in neurons. That said, the manuscript does require a few alterations before publication. We outline substantive comments below*.

*1) A key negative result is the lack of significant change in EBP tracks after axon injury in* efa-6 *mutants (*Figure 5*). Given the importance of this negative result to the model, the power of the experiment should be discussed. Ideally it would be nice to see some independent confirmation, perhaps with another allele*.

As the reviewers pointed out, this is a critical negative result. Calculations of statistical power using standard methods (e.g. http://powerandsamplesize.com/Calculators/,
www.ai-therapy.com/psychology-statistics/) indicate that effects of >15% should be detectable in experiments of this sample size (n > 10 per genotype). We have added a note on the statistical power of these experiments in the Discussion (see subsection “EFA-6 and injury-triggered MT dynamics”).

We have now also tested an independent deletion allele of *efa-6*, *ju1200*, for microtubule dynamics (EBP tracks) after axon injury. In the EBP::GFP assay *ju1200* resembles the *tm3124* allele (Figure 5—figure supplement 1), consistent with its similar phenotypes in axon development and regeneration (Figure 2).

*2)*
Figure 7—figure supplement 1*, part A has a cartoon illustrating quantification of co-localization. However, there is no formal description defining what exactly is being measured by F*^*in*^
*and F*^*out*^*, or how the F*^*in*^*/*^*out*^
*ratio is calculated. The Results section refers to the Methods, but the Methods section does not really describe the details. The authors should specify the details of how this ratio is calculated*.

We now provide more details on the calculation of this ratio in the Methods section (see subsection “Live imaging, laser axotomy, and FRAP”).

*3) The authors perform a really nice touch neuron-specific knockout experiment for TAC-1 using a single-copy, MOSCI-based, floxed, wild-type TAC-1 transgene (called* juSi162*) and a transgene that expresses Cre recombinase solely in the touch neurons. They have a PCR strategy for verifying that the recombination occurs as described (found in*
Figure 4—figure supplement 1*). However, the specifics of the PCR assay and the primers are not described in the paper. It's unclear if the worms from panel E also contain the* tac-1(ok3305) *deletion mutation. Presumably they do, and that the middle black primer falls within the deletion such that no PCR products are from the endogenous gene? The authors should state this and describe the experiment in more detail*.

We have added detailed information about the strains and PCR primers to the legend of Figure 4—figure supplement 1.

*4)*
Figure 6
*is difficult for the reader to navigate back and forth from the main figure and the supplement. More importantly, there are interesting and important results in the supplement that really should be in the main Figure (i.e., that injury does not alter ZYG-8 subcellular localization, and that TAC-1 and EFA-6 subcellular localization are not dependent on one another). The authors should move the panels A, B, and E from*
Figure 6—figure supplement 1
*into the main body of*
Figure 6.

We agree with the reviewers and have re-arranged the figures accordingly.

*5)*
Figure 5*. The authors do not observe a change in MT plus end growth in* tac-1(lf) *mutants in the graph under panel*
***A****. Presumably this experiment was done with the* ts *allele? What happens in the deletion allele examined with the Cre recombinase approach? This later genotype is more likely to be a null. This experiment could address whether or not TAC-1 is required for the EFA-6-mediated changes in MT dynamics (similar to what is observed for ZYG-8). Also, for the graph in panel B*, *why don't the authors show the data for the* efa-6(lf)*;* zyg-8(lf) *and* efa-6(lf)*;* tac-1(lf) *double mutants?*

The analysis of MT plus end growth in *tac-1(lf)* in Figure 5 was done using the *ts* allele *or455*. As the reviewer suggested, we have now examined MT growth in the *tac-1* deletion mutant *ok3305* (see subsection “EFA-6 is required for the injury-induced downregulation of axonal MT dynamics”). We rescued the lethality of *ok3305* with a single copy transgene containing a lox-flanked *tac-1* genomic cassette, and deleted the transgenic wild type *tac-1* copy in touch neurons using a tissue-specific Cre transgene. We observed a significant reduction in MT growth in this tissue-specific *tac-1* null mutant. We have included the data in Figure 5—figure supplement 1. We did not integrate these data into Figure 5 as different control strains were used.

As the reviewer requested, we have now added 3 hr MT growth data for *efa-6(lf); zyg-8(lf)* and *efa-6(lf); tac-1(lf)* double mutants to Figure 5
*efa-6(lf)* was not able to rescue the defect of *tac-1(lf)* or *zyg-8(lf)* in MT dynamics, suggesting they function in the same pathway.

*6) The authors identified ZYG-8 and TAC-1 as EFA-6 binding partners through a yeast two-hybrid screen. They then confirmed the interaction using GST-pull downs of proteins expressed in HEK293 cells. If this were a study being conducted in mammalian brain, one might suggest that they verify whether such protein complexes* can *be isolated between endogenous proteins from intact brain. However, that would be an unreasonable request given that the* C. elegans *nervous system comprises such a small percentage of the total biomass of the animal and there is no way to dissect out large quantities of worm neurons for such biochemical studies. Such an experimental requirement should not be requested of these authors, and the significance of their* in vivo *genetic findings far outweigh the absence of data supporting protein–protein interactions of the endogenous proteins*.

We appreciate the reviewers’ understanding that such biochemical studies are technically challenging. We are currently pursuing proteomic approaches to identify EFA-6 interactors in *C. elegans* neurons, however such experiments are still in their early stages.